# Restaurants Offering Healthier Kids’ Menus: A Mixed-Methods Study

**DOI:** 10.3390/nu17101639

**Published:** 2025-05-10

**Authors:** Tim A. van Kuppeveld, Bernadette J. Janssen, Kirsten E. Bevelander

**Affiliations:** 1Sociology, Faculty of Social Science, Radboud University, Thomas van Aquinostraat 4, 6525 GD Nijmegen, The Netherlands; 2Municipal Health Service GGD Gelderland-Zuid, Professor Bellefroidstraat 22, 6525 AE Nijmegen, The Netherlands; 3Primary and Community Care, Radboud University Medical Center, Geert Grooteplein 21, 6525 EZ Nijmegen, The Netherlands

**Keywords:** restaurant meals, children, healthier kids’ menus, motivations, obesogenic environment

## Abstract

Introduction: The food environment is an important determinant of children’s eating behavior. Improving the environment to encourage healthier choices is crucial to prevent obesity, especially in restaurants where the majority of kids’ menus are unhealthy. This study explored the perceptions, attitudes, motivations, influencing factors, and opportunities of restaurant owners, managers, and chefs for implementing healthier kids’ menus in Dutch restaurants. Method: We used a mixed methods design in two consecutive study parts. Part I consisted of an online unstandardized questionnaire that was completed by 44 restaurant owners, 26 chefs, 18 managers, and 6 other restaurant employees (n = 94). This was followed by semi-structured interviews with 3 restaurant owners, 2 chefs, and 1 manager, to gather exploratory information in Part II (n = 6). The quantitative data were categorized into three groups: restaurants without kids’ menus (n = 18), restaurants with unhealthy kids’ menus (n = 24), and restaurants with (partially) healthy kids’ menus (n = 52). Group differences were assessed using the Kruskal–Wallis test. We used thematic analysis for the interviews. Results: Parts I and II showed that the restaurant sector is aware of the need, and willing and motivated to offer healthier kids’ menus. Nevertheless, the concerns about food waste, the unhealthy demand from children and parents, and seeing eating out as a free pass to consume unhealthy meals by children and parents were important factors limiting the implementation of healthier kids’ menus. Discussion: We discussed potential solutions to enhance demand and acceptance of healthier kids’ menus, such as attractive names, storytelling, offering children’s portions based on adult menus, and using participatory approaches in which parents, children, and chefs co-create meal composition.

## 1. Introduction

The worldwide prevalence of childhood obesity has increased dramatically over the past four decades and emerged as one of the most serious public health concerns in the 21st century [1,2,3]. Children with obesity are prone to become obese during adulthood and are more at risk of developing chronic diseases such as heart failure, stroke, type 2 diabetes, several types of cancer, and osteoarthritis [4,5]. Nowadays, approaches to reduce obesity have been increasingly focused on the obesogenic environment apart from the individual [6,7]. An obesogenic environment is defined as ‘the sum of influences that the surroundings, opportunities, or conditions of life have on promoting obesity in individuals or populations’ [8,9,10,11]. In Western societies, the current food environment is characterized by the widespread availability of inexpensive, calorie-dense foods and beverages, along with numerous external cues that trigger excessive consumption [12,13,14]. The food environment is an important determinant of children’s eating behavior [15,16,17]; therefore, improving the food environment to encourage healthier choices is an important aspect of obesity prevention [18]—for example, by promoting restaurants to offer healthier choices. Therefore, this study explored what motivates the restaurant sector to implement healthier kids’ meals.

The number of families getting their daily calories from food consumed outside the home is increasing, while restaurant meals are often more caloric, served in larger portions, and less nutritious than meals prepared at home [3,19,20,21,22]. Moreover, the majority of restaurants offer kids’ meals that lack vegetables and are high in calories, salt, and saturated fat [23]. Indeed, consumption of restaurant meals is associated with higher daily energy intake and poorer nutritional quality among children compared with dietary intake on days restaurants were not visited [24]. Restaurants offer a valuable opportunity to introduce healthier food options to children, potentially influencing their dietary preferences toward more nutritious selections [25]. Achieving a shift toward making healthier choices more self-evident for children dining in restaurants requires active collaboration from restaurant owners.

The Retail Food Environment and Customer Interaction Model (RECIM) [26], emphasizes a dynamic interplay between retailer-level factors (e.g., business operations, profit margins of restaurant owners), consumer behavior (e.g., purchasing choices, preferences of parents and children), and broader contextual influences (e.g., food culture, policy, media) to achieve a shift toward implementing healthier kids’ menus. These interconnected elements shape food choices at the point of purchase and influence how and whether healthier kids’ menus are implemented in restaurants. While scientific research is scarce on restaurant owners’ motivations for implementing healthier kids’ menus, existing studies indicate that personal commitment to promoting healthy lifestyles [27,28] and the potential for positive (media) attention when offering healthier food alternatives [27] can play a role. Importantly, a study showed that restaurant owners believe that a reduction in salt and fat content in kids’ meals is easy to implement [29]. For example, by offering healthy and varied meals from the adult menu in smaller portions for children [24]. Another study showed a shift in future chefs’ priorities, focusing not only on taste but on using healthy ingredients when preparing food in restaurants [30]. These findings highlight the importance of intrinsic motivations that restaurant owners and chefs should have to create meals that are both tasty and healthy [30].

Importantly, the perceived consumer demand for healthier kids’ menus seems to be a crucial consideration for restaurant owners. Although research indicates that parents and children are open to healthier kids’ menus (e.g., healthier preparation methods, smaller portions, or fruits and vegetables as side dishes) [31], and that consumers generally express a willingness to pay more for healthier food [32], these intentions are often not reflected in actual purchasing behavior when dining out [28,33]. This has been explained by the belief that eating at a restaurant is a special occasion where unhealthy food consumption is allowed [24,27,34]. In addition, parents do not want to be bothered by children’s food pickiness when offered vegetables, especially when eating out [35].

From the restaurant owners’ perspective, this weak consumer signal, combined with operational challenges and financial risks, makes implementation difficult. Restaurant owners express concerns about the extent to which children and parents embrace and accept healthier choices on the menu and the related negative financial consequences [24,28]. They perceive menu changes as an extensive task and a potential risk to profits [24,36], for example, because vegetables, fruits, and other fresh healthy products often have a limited shelf life and go to waste when not ordered [28,33,34]. Additional operational barriers include a lack of time, limited kitchen space, lack of healthy recipes, and insufficient staff knowledge or culinary skills [33,34].

To reshape the obesogenic environment, integrated approaches in public health promotion have emerged involving the expertise of multiple stakeholders [37]. It is essential to identify the factors that need to be addressed by these involved stakeholders to change behaviors and create systemic change. The restaurant sector can directly impact health promotion strategies aimed at reducing energy content in restaurants, thereby contributing to a decrease in children’s energy intake [20,38]. Nevertheless, research is needed to understand the motivations of the restaurant sector to ensure their cooperation in public health interventions. This research area has limited existing literature, and in the Netherlands, there is currently no study exploring the perceptions of restaurant owners, managers, and chefs regarding the implementation of healthier kids’ menus. Based on RECIM and previous literature, we explored the perceptions, attitudes, motivations, influencing factors, and opportunities of Dutch restaurant owners, managers, and chefs for implementing healthier kids’ menus [26].

## 2. Materials and Methods

### 2.1. Study Design

We used a mixed methods approach in two consecutive study parts in the restaurant sector. In Part I, an online questionnaire was distributed to collect information about motivations for the composition of current kids’ menus, and future motivations, attitudes, and influencing factors to offer healthier kids’ menus. In Part II, semi-structured interviews were conducted to elaborate on the findings of Part I. Inclusion criteria were that participants worked in restaurants that had at least one certified chef in the kitchen and that children could also eat in the restaurant. Owners and employees of cafeterias, snack bars, and sports canteens were excluded because they do not employ certified chefs or offer freshly prepared meals. This study was conducted in accordance with the Declaration of Helsinki and the GDPR regulations and was approved by the Ethics Committee Social Science of Radboud University (ECSW-2019-090, 6 June 2019). All participants provided active consent.

### 2.2. Study Setting

In 2019, the first local prevention agreement (LPA) was signed in the Dutch city Nijmegen, initiated by the movement ‘Green, Healthy and on the Move’ [39,40]. It has grown to a network consisting of more than 70 local organisations that aim to promote a healthy lifestyle among employees and citizens. The network facilitates organizations to exchange knowledge and best practices, and set up collaborations. This study was facilitated in light of an intervention program with an integrated approach, developed in collaboration by multiple LPA partners, comprising an academic institution, the municipal health service, local restaurants, elementary schools, and the children’s council, to promote healthy kids’ meals in restaurants in Nijmegen. Primary school children and chefs visited each other at school and restaurants to co-create healthier kids’ meals, which were placed on the menu at the restaurants [41].

### 2.3. Study Sample Part I

The participants were recruited via national Facebook groups for restaurant employees and the (social media) network of ‘Green, Healthy and on the Move’. In the period from April to May 2022, restaurant employees were asked to complete the questionnaire, administered on an accessible digital research platform for health research [42]. The study took approximately 10–15 min, and data were collected anonymously.

Participants who did not finish the questionnaire were included when they progressed beyond the provision of background information and the first 10 items. This decision was made in light of the challenges encountered in reaching the target population, and every response received was deemed valuable for the study. Out of the initial 156 participants who started the questionnaire, 94 participants were included, consisting of 44 restaurant owners, 26 chefs, 18 managers, and 6 other employees of restaurants in the Netherlands. Participants described the restaurants where they work as luxury restaurants (23.6%), family restaurants (22.6%), or café-restaurants (20.8%). Also, grand cafés (10.4%), bistros (8.5%), lunch cafés (4.7%), roadside restaurants (0.9%), and other types of restaurants (8.5%) were named. Most restaurants had at least one certified chef in the restaurant (91.1%).

### 2.4. Measures Part I

Our questionnaire was constructed based on topics found in the existing literature. No validated questionnaires were available specifically addressing motivations for offering kids’ menus.

#### 2.4.1. Current Kids’ Menus

Participants were asked to rate their perception of the healthiness of the kids’ menus they offered using the options ‘unhealthy’, ‘partially healthy’, or ‘healthy’, and ‘none’ if they did not offer kids’ menus. Additionally, participants offering kids’ menus were asked to rate the healthiness of the current kids’ menu on a 10-point Likert scale, ranging from ‘not healthy’ (1) to ‘very healthy’ (10).

#### 2.4.2. Motivations for Menu Composition

We examined motivations for the composition of the current kids’ menus and the consideration of offering a healthier kids’ menu in the future using a multiple-choice question format based on previous research [24,27,28,29,43]. Answers included, for example, ‘to meet the demand for healthy kids’ meals’, ‘because it is a special occasion’, and ‘to offer new flavors to children’. In addition, an open response question provided room for an alternative explanation for the composition of the kids’ menu. Participants who offered unhealthy kids’ menus were asked if they had ever tried to offer a healthy kids’ menu. Additionally, the main reasons for not offering a healthy kids’ menu (anymore) were asked with an open-ended question.

#### 2.4.3. Menu Change Options

Eleven multiple-choice options were presented with the aim of improving the kids’ menu with healthier options [31,44]. For example, answer options were ‘use less processed products’, ‘use less fried products’, and ‘offer smaller portions’.

#### 2.4.4. Attitudes Toward Menu Changes

The necessity, willingness, and ease of change to offer a healthier kids’ menu were asked on three 10-point Likert scales [24,29]: (1) ‘To what extent do you think it is necessary to offer a healthier kids’ meal?’ (1 = ‘not necessary’ to 10 = ‘very necessary’), (2) ‘To what extent are you willing to offer a healthier kids’ meal?’ (1 = ‘not willing’ to 10 = ‘very willing’), and (3) ‘To what extent do you think making changes toward a healthier kids’ menu is easy?’ (1 = ‘very difficult’ to 10 = ‘very easy’).

#### 2.4.5. Factors Influencing Change

Participants were provided 28 statements derived from the literature, each addressing factors that influence change in the provision of a healthier kids’ menu [10,24,27,28,36,44,45]. The statements addressed finance, ingredients, changes, and needs. These statements could be answered with ‘completely disagree’ (1) to ‘completely agree’ (4). Furthermore, to explore what kind of support participants needed to change their menus into healthier kids’ menus, a question with multiple-choice answers was used. For example, answering options were ‘access to more healthier kids’ meal recipes’, ‘simple strategies to make kids’ menus healthier’, and ‘proof that a healthier kids’ menu is profitable’ [10,45]. In addition, an open response question provided room for alternative needs to make a change to a healthier kids’ menu.

### 2.5. Analysis

To address the variation in the availability of healthier kids’ menus among restaurants and potential differences in perceptions, we categorized the restaurants into three distinct groups: restaurants with no kids’ menus (n = 18), restaurants with unhealthy kids’ menus (n = 24), and restaurants with (partially) healthy kids’ menus (n = 52). Participants with ‘partially healthy kids’ menus’ (n = 48) and ‘healthy kids’ menus only’ (n = 4) were pooled due to the small number of participants with healthy kids’ menus only.

Means and standard deviations were used to measure the health rating of the current kids’ menus. The Kruskal–Wallis test was used to calculate significant differences between restaurants with unhealthy and (partially) healthy kids’ menus due to the unequal distribution between the groups. Frequencies were calculated for the motivations for the composition of the current menus, and future motivations, menu change options, influencing factors, and opportunities to offer a healthier kids’ menu. Percentages are higher than 100% because this information was collected using multiple-choice questions, where participants could choose all the answers that applied. Open responses for reasons not offering healthy kids’ menus were categorized. Means and standard deviations were used to measure the necessity, ease of change, willingness, and influencing factors to offer healthier kids’ menus. Significant differences between the three groups were calculated using the Kruskal–Wallis test. To determine which groups were significantly different, a post-hoc test with Bonferroni correction was used. As participants had to answer a minimum number of questions to remain in the sample, questions later in the questionnaire were not answered by the full sample. The data were analyzed using IBM SPSS Statistics version 29 and Microsoft Excel version 16.

## 3. Results Part I

### 3.1. Current Kids’ Menus

Restaurants offering unhealthy kids’ menus provided significantly lower health ratings for their current kids’ menus compared to those with (partially) healthy kids’ menus (*H*(1) = 33.59, *p* = 0.001; see Table 1).

### 3.2. Motivations for Menu Composition

The two main motivations behind the current menu compositions in restaurants offering (partially) healthy and unhealthy kids’ menus were the demand for either healthy or unhealthy kids’ meals from children and parents, and the perception of dining out as a special occasion (see Table A1). Restaurants that served healthy menus thought carefully about it and were also motivated to offer children new flavors, in contrast to those providing unhealthy kids’ menus.

All three restaurant groups were increasingly motivated to improve kids’ menus, driven by a rising demand for healthier options (49.4%). Key drivers were promoting innovation and originality in kids’ meals (48.1%), enhancing the restaurant’s public image (46.9%), and supporting a healthier lifestyle for families (43.2%). Additionally, many aim to introduce children to new flavors (39.5%) and make a meaningful contribution to children’s overall health (29.6%; see Table 2). Restaurants without kids’ menus, in particular, were motivated to contribute to a healthier food environment (60.0%). Open-ended responses emphasized the importance of offering healthy kids’ meals that are not only nutritious but also tasteful and appealing to children. Furthermore, some participants expressed a desire to align kids’ menus with current healthy living trends and to ensure that parents are informed about the availability of healthy options for their children in restaurants.

### 3.3. Menu Change Options

According to the participants, the most important contributors to creating healthier kids’ menus include reducing the presence of processed items (59.6%), minimizing fried products (59.6%), lowering salt content (47.9%), and decreasing added sugars (46.8%; as shown in Table 3).

### 3.4. Attitudes Toward Menu Changes

There were significant differences in attitudes regarding the ease of establishing a transition to healthier kids’ menus (*p* = 0.042), especially between restaurants with no kids’ menus and those with unhealthy kids’ menus (*z* = 2.51, *p* = 0.036; see Table 1). There were no significant differences among the three groups in terms of willingness (*p* = 0.627) and recognition of the need (*p* = 0.170) to offer healthier kids’ menus, with all three groups expressing a high level of willingness (*M* = 7.3) and awareness of the need (*M* = 6.8) to make this change (see Table 1).

### 3.5. Factors Influencing Change

Similar to the findings on motivations, participants indicated demand as the most important factor of change to serve healthier kids’ menus. As indicated in Table 4, the perceived lack of demand for healthy kids’ meals among children was reported by 87.5% of the participants. There was no significant difference in this perception between the three restaurant groups (*p* = 0.083). Additionally, restaurants commonly perceived that children frequently do not consume fruits and vegetables included in kids’ meals (71.2%, *p* = 0.216). In open-ended responses, this was mentioned as the main reason for the removal of healthy kids’ meals from the menu (n = 9).

In contrast to restaurants with (partially) healthy kids’ menus (67.6%), restaurants with unhealthy kids’ menus (93.8%) perceived low parental demand for healthy kids’ meals as a significant barrier (*z* = 2.49, *p* = 0.038). According to all three restaurant types, parents frequently play a role in selecting unhealthy food options for their children when eating out (96.8%, *p* = 0.839). In addition, all three restaurant groups have a high level of skepticism regarding the understanding and acceptance of healthier kids’ menus by children (79.7%, *p* = 0.220) and parents (62.5%, *p* = 0.366). Therefore, evidence demonstrating that children and parents desire healthy kids’ meals is crucial to motivate restaurants to include healthier options on their menus (71.0%, see Table A2).

Restaurants with unhealthy kids’ menus found it significantly more time-consuming and labor-intensive to create healthier options than restaurants with (partially) healthy kids’ menus (*z* = 2.37, *p* = 0.018) or no kids’ menus (*z* = −2.82, *p* = 0.005). Additionally, these restaurants were significantly more likely to base their kids’ menus on popular meals that lack healthy options, compared to restaurants with (partially) healthy kids’ menus (*z* = 2.44, *p* = 0.015) and restaurants with no kids’ menus (*z* = −2.45, *p* = 0.014). Furthermore, compared to restaurants with no kids’ menus, restaurants with unhealthy kids’ menus were significantly more likely to report having too few recipes for healthy kids’ meals (*z* = −2.74, *p* = 0.020; see Table 4). Finally, the results showed that 21% of the restaurants are in need of more recipes for healthy kids’ meals (see Table A2). Table 4 also shows that a large number of restaurants developed kids’ meals based on what is available on the adult menu (57.6%, *p* = 0.192).

### 3.6. Conclusion Part I

The results of Part I highlighted the important impact of children’s and parents’ demand, whether for healthy or unhealthy kids’ menus, on both the motivations for current compositions, future motivations, and influencing factors to introduce healthier options in restaurants. Within the restaurant sector and regardless of their current kids’ menu, there is willingness and awareness to offer healthier kids’ menus, along with a belief that the transition will be relatively uncomplicated. To bring about change, restaurants need evidence that children and parents desire healthier kids’ meals, as well as inspiration for how to implement those meals. In Part II, we explored explanations for these findings.


## 4. Methods Part II

### 4.1. Study Sample

In May 2022, individual semi-structured interviews took place with 3 owners, 2 chefs, and 1 manager (n = 6) of restaurants in the network of ‘Green, Healthy and on the Move’. The participants were recruited through telephone and e-mail contact. The restaurants where they were employed often served as diners and lunch locations, and they were situated in either neighborhood settings (n = 4) or near/in the city center (n = 2).

### 4.2. Procedure

The interview guide was based on literature and insights from Part I. The interview guide included topics concerning participants’ perceptions and experiences regarding the nutritional quality of the current kids’ menu, and the motivations and factors influencing the provision of a healthier kids’ menu (see Table A3). The pilot testing of the interview guide resulted in minor adjustments to improve clarity. The interviews were conducted in person or by telephone by the main author (T.A.v.K) and lasted approximately 30 min. The interviews were recorded and transcribed verbatim. The transcripts were pseudonymized before analysis. We aimed to conduct 5–7 interviews due to the exploratory nature of the study.

### 4.3. Analysis

Data were analyzed thematically [46], supported by software ATLAS.ti (version 23.0.7.0). First, all transcripts were read to gain familiarity with the data, and one author (T.A.v.K.) applied open coding to three transcripts. The relevant codes and quotes were identified based on the aim of this study and discussed between two authors until consensus was reached (T.A.v.K., and K.E.B.). The scope and names of the codes were discussed, resulting in a first conceptual coding structure. This conceptual coding structure was applied to the next three transcripts by the main author (T.A.v.K.). The list of quotes and codes from this step was discussed again (T.A.v.K., and K.E.B.), after which similar and duplicate codes were merged, resulting in a final coding structure. This final coding structure was systematically applied to all transcripts by the main author (T.A.v.K.) until all interviews were coded.

Next, the codes were grouped into broader categories and discussed among two authors (T.A.v.K. and K.E.B.) until consensus was reached. For example, codes concerning motivations to offer healthier kids’ menus (e.g., appropriate to the concept of the restaurant, contribute to a healthier society, teach children new tastes, and have a goal for healthier eating) were grouped in the code ‘motivations: offering healthier kids’ menu’. Groups of codes that linked to shared concepts were then identified and discussed, such as needs and opportunities to offer healthier kids’ menus. This analysis phase resulted in four main themes: (1) meal preparation, (2) commercial interests, (3) parental influence, and (4) implementation strategies. The subthemes within these are described in the results section. The overall results were shown in Table 5. Quotes were translated from Dutch to English and used to illustrate the findings.

## 5. Results Part II

### 5.1. Meal Preparation

Preparing healthier kids’ menus posed several practical challenges in the kitchen. These challenges included the time-intensive preparation of fresh ingredients compared to the convenience of using frozen ingredients. Participants mentioned that they often used frozen products due to their convenience, extended shelf life, affordability, and potential to reduce food waste. Additionally, they are relatively easy to prepare and often consist of items such as fries, pizza, and pancakes. Nevertheless, they also acknowledged that using frozen products does not entirely mitigate the issue of food waste, especially concerning children, who frequently do not consume the entire portion. Ensuring suitable portion sizes for children was mentioned as another important challenge, primarily due to the considerable variability in energy requirements and food preferences within this age group. The following quotes illustrate this:


*“We keep the normal portion size of the salad, because at least then we give the impression that we are offering it. Suppose you take it off, then parents who want their children to eat that no longer have the opportunity to do that. So, we always do it, well then often we have to throw it away. Yes, that’s unfortunate, but it does happen”.*
[Restaurant manager 1] 


*“I think it’s a shame, but it’s still difficult. You don’t want to give too little, because people pay for it. You get that back. But when is something too little or too much? One child is not the other, of course”.*
[Chef 1] 

### 5.2. Commercial Interest

Commercial interest comprised barriers and motivations. Reasons mentioned for not offering kids’ menus included a perceived lack of demand, infrequent sales, and limited space on the menu for text. In general, participants acknowledged that there was a strong preference among children for easier, less healthy options, with items like fries and snacks being the most popular, while healthier kids’ meals were rarely chosen. Consequently, there was little to no interest or perceived need to provide healthier options for children, which led to a predominant focus on offering mainly unhealthy kids’ meals in restaurants.


*“If you find that 9 out of 10 children take a grilled cheese sandwich and 1 person chooses a slightly healthier meal, which you should stock specifically for that child. Then you have no use for it. It actually only costs you money. It is not profitable”.*
[Restaurant owner 3] 

Still, participants saw commercial value when the healthier kids’ meals fitted into the restaurant’s concept. Moreover, introducing children to new flavors and responding to the healthier preferences of children and parents was regarded as a way to contribute to a healthier society, and a nice, enjoyable, and meaningful endeavor.


*“Look, if you then get a chance here to turn that a little bit and if you can make tasty meals out of that, yeah then as far as I am concerned, you are just doing good for people”.*
[Chef 2] 

Furthermore, participants mentioned a growing trend in the demand for and availability of vegetarian and organic options in restaurants. They also suggested that the location of restaurants within a neighborhood can influence supply and demand dynamics. In particular, young families with children were identified as a target group open to healthier kids’ menus/meals.


*“I think because we’re pretty much in a thriving neighborhood, all young families with kids, and I think as Nijmegen, as a green city, that it is quite alive here in the neighborhood and can appreciate it”.*
[Chef 1] 

To improve the healthiness of kids’ meals, a number of suggestions were made, such as including a greater variety of fruit and vegetables, reducing the presence of sugary drinks, and using healthier cooking methods when preparing meals.


*“If you look at what a child needs, well they don’t need trays of salt or sugar, and don’t need flavor enhancers or e-numbers. So, if one leaves those out now, then at the base you have already gained a lot in creating a responsible kids’ meal”.*
[Restaurant owner 1] 

### 5.3. Parental Influence

The influence of parents’ associations with eating out and unhealthy food played an important role in their children’s unhealthy food choices. According to the participants, parents often make the decision to choose unhealthy foods on behalf of their children. This may partly be due to parents’ desire for a trouble-free evening with minimal resistance from their children. Participants recognized that parental behaviors, including a preference for convenience and relaxation, contribute to the prevalence of unhealthy choices for children in out-of-home dining settings. Additionally, eating out was perceived as a special occasion where children are treated to unhealthy options such as fries and snacks. This view contributes to the prevalence of unhealthy choices in restaurants.


*“The moment you have to have the conversation with the child and the child indicates they don’t like it, then you’re more concerned with that and I think that’s a part that parents don’t feel like doing when they’re out to dinner”.*
[Restaurant owner 1] 


*“Because a child gets to enjoy a moment, because we are out to eat, it is a festive day when you eat out. I think that’s the biggest stumbling block to switching to a healthier kids’ menu”.*
[Restaurant manager 1] 

On the other hand, participants indicated that parents sometimes appreciate the availability of healthier kids’ meals. When parents are more aware of healthy eating, they are more likely to appreciate and seek out healthier options for their children when eating out.


*“If the parent chooses a vegan batter or a pancake without animal products, you see that it gets copied to the kids and they get a vegan pancake too”.*
[Restaurant owner 2] 

### 5.4. Implementation Strategies

Despite the commercial and social challenges, restaurant strategies were identified for implementing healthier kids’ menus. The strategies included creativity, kids’ meals based on adult menus, and parental involvement. Participants consistently emphasized that creativity in meal presentation is important, as well as having access to resources to create healthy and tasty meals for children. They believed in the importance of translating children’s preferences and enjoyment into well-developed, healthier kids’ menus to meet their needs while encouraging healthier choices. Several suggestions were made for making healthier kids’ menus more appealing to children. These included creating appealing names and background stories behind the meals or providing special cutlery and gifts for children.


*“We want to be right in the sweet spot, so that it’s indulgent and healthy at the same time. And finding that sweet spot is actually the big challenge of a healthier kids’ menu”.*
[Restaurant manager 1] 

Also, the local health organization’s program in the city of Nijmegen was mentioned, in which children worked with chefs to develop new, healthier kids’ meals, resulting in healthier kids’ meals on the menu. The children’s ideas for healthier menu options were highly valued by the participants and were used to guide menu changes. As a result, some participants expressed their intention to participate in the local program again.


*“You notice that they come up with things we wouldn’t think of ourselves. That we quickly don’t do something because the children don’t like it”.*
[Restaurant owner 2] 

In addition, participants mentioned the need to balance healthy and unhealthy meal options in kids’ menus to avoid excluding people. This approach aimed to meet children’s different needs and preferences. Another strategy to deal with infrequent sales and lack of demand was basing kids’ meals on the adult menu. With this approach, the same meals and preparations are used for all dinners, but with smaller portions served to children. This approach not only streamlined the children’s dining experience but also encouraged them to develop healthier eating habits, taste development, and taste broading when dining out.


*“We try to encourage the children to eat with the parents and just eat a normal meal, with an appropriate portion of course”.*
[Chef 2] 

Participants who based kids’ meals on ingredients already available on the adult menu often reported consulting with parents and children about the meal options. Through this collaborative approach, the customer’s preferences as well as the culinary resources and capabilities of the kitchen can be taken into account. For example, one approach mentioned was that kids’ meals were not explicitly mentioned on the printed menu anymore. Instead, verbal explanations and details of kids’ meal options were provided when specifically requested by parents.


*“I usually throw up a few balls in consultation with the parents, and then you soon notice by the child’s reactions what the children like and enjoy”.*
[Restaurant owner 1] 

### 5.5. Conclusion Part II

Part II identified several factors influencing the implementation of healthier kids’ menus. Practical challenges included the increased time required for fresh ingredient preparation and difficulties in ensuring appropriate portion sizes for children. Commercial interest was hindered by low demand, infrequent sales, and limited menu space, although a growing trend for vegetarian and organic options among young families was noted. Parental influence played a dual role by contributing to both unhealthy choices and an increased openness toward healthier options. Finally, innovative implementation strategies, such as creative presentation, integration with adult menus, and parental consultation, were highlighted as essential for facilitating change.

## 6. Discussion

The current study was the first to explore the perceptions, attitudes, motivations, influencing factors, and opportunities of restaurant owners, managers, and chefs for implementing healthier kids’ menus in Dutch restaurants. Our findings show that the restaurant sector is aware of the need, and willing and motivated to offer healthier kids’ menus. Nevertheless, the concerns about food waste, unhealthy demand, and seeing eating out as a free pass to consume unhealthy meals by children and parents were important factors limiting the implementation of healthier kids’ menus.

In general, our study showed that regardless of restaurant type, all restaurants struggled with the same challenges to implement healthier kids’ menus. Especially, restaurants offering primarily unhealthy kids’ menus regarded this as difficult and time-consuming. A previous study also showed that restaurants often describe customizing kids’ menus as a time-consuming, iterative process with multiple steps that yield little profit [24]. Moreover, a crucial factor limiting the implementation of healthier kids’ menus is the demand and preference for unhealthy options among both children and parents. For example, fries with a snack were most popular, while healthier kids’ meals were rarely chosen. In line with previous research, we found that restaurants are discouraged from offering healthy kids’ menus because they are not convinced that these healthier options are embraced by parents and children [24]. This was mainly explained by the fact that children and parents perceive eating out as a moment of indulgence [27,43]. Additionally, restaurant owners see this as a potential risk to profits [33,34]. The main motivation for restaurants with (partially) healthy kids’ menus to offer healthier kids’ menus was indeed to meet the demand of children and parents for healthy kids’ meals. These findings align with the Retail Food Environment and Customer Interaction Model (RECIM) [26], suggesting that the shift toward implementing healthier kids’ menus is perceived as difficult due to the financial and operational risks.

Previous studies have shown that both children and parents are open to changes in kids’ menus, including healthier preparation methods, smaller portions, healthier drink options, and the inclusion of fruits and vegetables as side meals [25,31], and that consumers generally express a willingness to pay more for healthier food [32]. In our study, restaurant owners pointed out that the location of restaurants within a neighborhood can affect these healthier demands. A previous study showed that restaurants located in low-education neighborhoods offered less healthy food compared to those in higher-education areas [47]. Consistent with RECIM [26], this lower availability of healthy foods in less educated neighborhoods could be due to low demand, though an inverse causality cannot be ruled out [48]. To facilitate a transition toward healthier kids’ menus in restaurants, it is imperative that restaurants are aware of the demand among children and parents. Therefore, a follow-up study could further explore consumers’ perceptions of eating out. It would be interesting to understand whether and how these perceptions can be changed over time, and whether there are differences based on consumer characteristics [26]. In addition, the extent to which restaurants are actually aware of parents’ and children’s expectations for healthy kids’ meals should be included.

Next to the customers’ demand, our study found other motivations to implement healthier kids’ menus, such as social responsibility, profiling, and innovativeness. In line with previous research, we found that restaurant owners wanted to contribute to a healthier society by responding to children’s unhealthy lifestyles. For example, by using less fried and processed products, salt, and added sugars. While RECIM does not offer room for social responsibility, the Corporate Social Responsibility Model (CSRM) does [49], emphasizing that ethical and moral considerations can significantly influence corporate decision-making. In this context, the role of restaurants in improving the nutritional value of kids’ meals [29,30] reflects a moral commitment to public health and social well-being, extending beyond purely economic or competitive interests. In addition, some restaurants are keen on following nutritional trends to remain competitive and respond to changes in consumer behavior [28,33]. Innovation in meal options for children can also contribute to creating a more positive restaurant profile [27]. This aligns with both RECIM and CSRM in recognizing how business models and product offerings are shaped by customer preferences, market positioning, and the pursuit of financial sustainability [26,49]. To implement healthier kids’ menus, restaurants identified several strategies. Possible solutions include increasing the attractiveness of healthy kids’ meals by using appealing names and stories, offering children’s portions based on adult menus, and using a participatory approach in which parents, children, and chefs decide on the kids’ meals. It is important for restaurants to tap into children’s creativity by brainstorming ideas for tasty and healthy kids’ meals. Furthermore, this study showed the opportunity of offering menu items from the adult menu in smaller portions for children. This approach is in line with findings from a previous study that also mentions the added benefit of reducing food waste by eliminating the need to buy separate ingredients and presenting healthier options as natural and normative to encourage healthy eating in children [50]. In addition, to achieve more significant improvements in children’s diets, health promotion efforts should focus on creating a family-based approach where healthier choices are readily available and accessible in restaurants [25]. Involving parents and families in initiatives to promote healthy eating habits in children is crucial because of the strong links between family food practices, parental intake, and child intake [51,52,53]. For example, a participatory approach, where the kids’ menu is no longer explicitly on the menu, but parents, children, and chefs decide on the kids’ meals together. In this study, almost all restaurants reported that parents are involved in choosing what children eat, which is often unhealthy. Restaurants stated that parents often make the final choice of unhealthy meals for children in order to have a relaxed evening in the restaurant without resistance. At the same time, consistent with a previous study, the current study found that restaurants believe it is the responsibility of parents to make appropriate food choices for children in restaurants [29]. Therefore, parent involvement in a participatory approach can be used positively to choose healthier options for kids’ meals. These approaches align with RECIM, which emphasizes the dynamic interplay between consumer preferences and food retail practices [26]. To engage the restaurant sector in health-promoting activities, incentives (e.g., subsidies or tax incentives) and community participation are needed to facilitate restaurants in changing their menus and overcoming their business-related concerns. A recent study substantiates the effectiveness of pricing policies, such as subsidies or tax incentives, in influencing consumer behavior by shifting purchases and consumption toward healthier options [54]. These strategies acknowledge how policy and economic context interact with retailer behavior and consumer choices to shape the food environment [26].

### Strengths and Limitations

This study had strengths and limitations. The strength of this study is that it was the first in the Netherlands in this area of research. In two consecutive study parts, a mixed methods approach was used to gain in-depth insights. This approach was necessary to collect relevant data in the absence of validated instruments for this specific area of research. A limitation of this study is the relatively small sample size that completed the questionnaire in Part I. Nevertheless, it was administered during the period in which various COVID-19 measures were announced, which caused restaurants to close and staff shortages [55]. These factors may explain the lower response rate to the questionnaire, making it necessary to include partially completed questionnaires in the sample. The relatively small sample size in Part I may have restricted the generalizability of our findings. Moreover, the contextual backdrop of pandemic-related disruptions could have influenced participants’ responses, introducing situational bias. However, each participation contributed to a better understanding in a novel research area. The insights will help to understand the motivations of the restaurant sector to ensure their cooperation in public health interventions. Given these limitations, we recommend that future research should further explore perceptions within the restaurant sector to implement healthier kids’ menus using a larger, more representative sample, in order to validate both the findings and the measurement instruments and identify temporal trends.

## 7. Conclusions

In conclusion, the current study showed that the restaurant sector is aware of the need, and willing and motivated to offer healthier kids’ menus. Nevertheless, concerns about food waste, unhealthy demand, and the perception of eating out as a free pass to consume unhealthy meals by children and parents were important factors limiting the implementation of healthier kids’ menus. To engage the restaurant sector in health-promoting activities, incentives (e.g., subsidies or tax incentives) and community participation are needed that facilitate restaurants to change their menus and overcome their business-related concerns. To implement healthier kids’ menus, possible solutions include increasing the attractiveness of healthy kids’ meals by using appealing names and stories, offering children’s portions based on adult menus, and using a participatory approach in which parents, children, and chefs decide on the kids’ meals. The Retail Food Environment and Customer Interaction Model [26] provides a model for the strategies that focus on improving customer interactions and the influence of consumer preferences in shaping product offerings, as well as the role of business practices in meeting these demands. This approach provides practical starting points for integrating healthier kids’ menus into current restaurant operations. Notably, social responsibility appears to play a role in this process and should therefore be considered within theoretical economic approaches.

## Figures and Tables

**Table 1 nutrients-17-01639-t001:** Attitudes toward current kids’ menus and offering healthier kids’ menus among restaurant groups with varying availability of healthy kids’ menus ^a^.

	**Total** **n = 94**	**(Partially) Healthy** **n = 52**	**Unhealthy** **n = 24**	**Without** **n = 18**	***p*-Value ^b,c^**
Health score of the current kids’ menu	5.5 (2.1)	6.4 (1.5)	3.5 (1.6)	-	0.001
Necessity to offer a healthier kids’ menu	6.8 (2.2)	6.9 (2.1)	6.3 (2.2)	7.1 (2.7)	0.170
Ease of changing to a healthier kids’ menu	7.1 (2.1)	7.0 (2.0)	6.3 (2.3)	8.1 (1.9)	0.042
Willingness to offer a healthier kids’ menu	7.3 (2.2)	7.3 (2.0)	7.2 (1.8)	7.2 (3.1)	0.627

^a^ Values are in means ± (SD). ^b^ The differences between the three categories were analyzed with the Kruskal–Wallis test. ^c^ The significance level is 0.050.

**Table 2 nutrients-17-01639-t002:** Future motivations to offer healthier kids’ menus among restaurant groups with varying availability of healthy kids’ menus ^a^.

	Total n = 81	(Partially) Healthy n = 45	Unhealthy n = 21	Without n = 15
Demand of children and parents for healthy kids’ meals	49.4	60.0	38.1	33.3
Innovations in choices	48.1	53.3	52.4	26.7
Positive attention for restaurants	46.9	48.9	52.4	33.3
Anticipate unhealthy lifestyles	43.2	40.0	52.4	40.0
Offer new flavors	39.5	44.4	33.3	33.3
Contributing to children’s health	29.6	28.9	28.6	33.3
Contributing to a healthier food environment	27.2	22.2	14.3	60.0
Creating a connection with guests	19.8	20.0	14.3	26.7
Following food trends	19.8	24.4	19.0	6.7
Comply with government measures	13.6	11.1	14.3	20.0
Preventing food waste	12.3	17.8	4.8	6.7
To remain competitive	6.2	6.7	4.8	6.7
Increase profits	4.9	8.9	0.0	0.0
More business opportunities	3.7	2.2	4.8	6.7

^a^ Values are in percentages. The sum of percentages is higher than 100% because participants could choose multiple answers that applied.

**Table 3 nutrients-17-01639-t003:** Opinion on contribution change opportunities for healthy kids’ menus among restaurant groups with varying availability of healthy kids’ menus ^a^.

	Total n = 94	(Partially) Healthy n = 52	Unhealthy n = 24	Without n = 18
Less fried products	59.6	63.5	62.5	44.4
Less processed products	59.6	65.4	33.3	77.8
Less salt	47.9	46.2	45.8	55.6
No added sugars	46.8	48.1	41.7	50.0
Smaller portions	42.6	38.5	37.5	61.1
At least 100 g of vegetables	41.5	40.4	45.8	38.9
Homemade sauces or healthier alternatives	38.3	46.2	16.7	44.4
Homemade spice mixtures	37.2	44.2	12.5	50.0
Less meat, fish, and poultry	21.3	23.1	16.7	22.2
Oil instead of butter in cooking	20.2	26.9	8.3	16.7
More whole grain variations	13.8	19.2	8.3	5.6

^a^ Values are in percentages. The sum of percentages is higher than 100% because participants could choose multiple answers that applied.

**Table 4 nutrients-17-01639-t004:** Factors influencing offering healthier kids’ menus among restaurant groups with varying availability of healthy kids’ menus ^a^.

	Total n = 65	(Partially) Healthy n = 38	Unhealthy n = 17	Without n = 10	*p*-Value ^b,c^
* Demand and needs*					
Parents often help decide on unhealthy choices	96.8	100.0	87.5	100.0	0.839
Low demand from children	87.5	83.8	94.1	90.0	0.083
Doubts about the understanding and acceptance of children	79.7	75.5	88.2	80.0	0.220
Low demand from parents	76.2	67.6	93.8	80.0	0.033
Doubts about the understanding and acceptance of parents	62.5	56.8	70.6	70.0	0.366
Does not belong to popular kids’ meals	47.6	40.5	75.0	30.0	0.014
Smaller portion sizes do not belong in restaurants	9.7	2.7	25.0	11.1	0.131
Healthy kids’ meals do not belong in restaurants	0.0	0.0	0.0	0.0	0.832
* Ingredients*					
Fruit and vegetables are often not eaten	71.2	71.1	82.4	54.5	0.216
Being developed based on the adult menu	57.6	47.4	64.7	81.8	0.192
Availability of healthy products is not always equal	34.8	44.7	23.5	18.2	0.266
The quality of healthy products is always equal	33.3	42.1	23.5	18.2	0.463
Pricey healthy products	31.8	31.6	41.2	18.2	0.220
Fruit and vegetables not long shelf life	27.3	26.3	35.3	18.2	0.351
Finance					
Contributes to sales	42.4	52.6	23.5	36.4	0.083
Provides financial benefit	40.8	51.3	31.6	23.1	0.444
Offered only when profitable	28.8	31.6	23.5	27.3	0.555
Important profit factor	14.7	21.1	10.5	0.0	0.408
Resources					
Lack of kitchen support	34.8	34.2	35.3	36.4	0.216
Lack of knowledge about healthy food among cooks	28.8	28.9	29.4	27.3	0.757
Lack of kitchen staff	28.8	23.7	47.1	18.2	0.090
Takes a lot of time and energy	27.7	21.1	47.1	20.0	0.007
Lack of knowledge about healthy preparation among cooks	27.3	31.6	23.5	18.2	0.365
Lack of healthy recipes	15.2	10.5	17.6	27.3	0.020
No clear approach	12.5	10.8	11.8	20.0	0.295
Does not fit the concept	10.6	7.9	11.8	18.2	0.136
No access to suppliers providing health products	6.2	5.3	5.9	10.0	0.197

^a^ Values are in percentages. The sum of percentages is higher than 100% because participants could choose multiple answers that applied. ^b^ The differences between the three categories were analyzed with the Kruskal–Wallis test. ^c^ The significance level is 0.050.

**Table 5 nutrients-17-01639-t005:** Overall results of Part II.

Main Themes	Sub Themes	Findings
Meal preparation	Practical challenges in preparation	- Preparing fresh meals is more time-consuming than using frozen products - Kitchen staff reported time pressure and limited capacity - Fresh preparations require additional planning and organization
	Preferences for frozen products	Frequently used due to - Convenience - Longer shelf life - Lower cost - Potential to reduce food waste - Commonly used items: fries, pizza, pancakes
	Challenges in preventing food waste	- Frozen products do not fully prevent food waste, especially with children - Children often do not finish their meals, even with small portions
	Challenges with portion sizes for children	- High variability in energy needs and food preferences among children - Difficult to offer a uniform portion size that meets individuals’ needs
Commercial interest	Perceived lack of demand	Many participants reported low interest in kids’ menus due to: - Infrequent sales - Perceived lack of demand from children and parents - Limited space on the menu - Unhealthy options (e.g., fries, snacks) were much more frequently chosen by children - Healthy kids’ meals were rarely ordered, leading to low perceived profitability
	Misalignment with commercial variability	- Stocking healthier meals was seen as costly and inefficient if rarely ordered - Low demand made healthier options economically unattractive
	Value when aligned with the restaurant’s concept	- Participants expressed interest when healthy meals matched the restaurant’s identity - Seen as a meaningful contribution to public health when done in a fitting, tasty way
	Anticipating changing preferences	- Chefs acknowledged the potential to introduce new flavors to children - Recognized a shift in parental and child preferences toward healthier meals
	Influence of the neighborhood	- Restaurants located in family-oriented or health-conscious neighborhoods saw more interest - Young families were identified as a promising target group
	Suggestions to improve healthiness	- Include more fruit and vegetables - Limit sugary drinks - Use healthier cooking methods and eliminate artificial additives
Parental influence	Parents often choose unhealthy options	- Parents frequently make the decision to order unhealthy meals on behalf of their children - Motivated by the desire for a smooth, conflict-free dining experience - Avoiding resistance or negotiation with the child is a key factor
	Dining out as a ‘special occasion’	- Eating out is often seen as a threat or celebration by parents - Unhealthy items like fries and snacks are considered part of the experience - This perception reinforces the prevalence of unhealthy kids’ menu options
	Parental habits and preferences	- Parents’ desire for convenience and relaxation shapes their children’s food choices - Children’s orders often reflect the habits and preferences of their parents
	Positive influence when health-aware	- Health-conscious parents appreciate the availability of healthier kids’ meals - Children may imitate their parents’ healthy choices (e.g., plant-based options) - Offering healthy alternatives can support and encourage healthier family eating behaviors
Implementation strategies	Creative presentation and experience	- Creativity in naming, storytelling, and presentation (e.g., special cutlery as gifts) helps attract children to healthy meals
	Collaboration with a local health initiative	- Participation in a local program where children co-create healthier meals was well received - Some participants expressed interest in participating again
	Balancing healthy and unhealthy options	- Offering both healthier and familiar indulgent items to avoid excluding preferences - Helps address the diverse needs of child diners
	Kids’ meals based on the adult menu	- Same meals as adults, but in smaller portions - More efficient for kitchen workflow - Supports children’s taste development and exposure to healthier choices
	Parent–children consultation at ordering	- Meal discussed with parents and children before ordering - Allows for customized options within kitchen capabilities - Sometimes kids’ meals are not listed but offered verbally on request

## Data Availability

The quantitative data are publicly available through the Radboud Data Repository.

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
