# Peer review of "Restaurants Offering Healthier Kids’ Menus: A Mixed-Methods Study"

_nutrients, 2025, doi:10.3390/nu17101639_

Round 1
Reviewer 1 Report
Comments and Suggestions for Authors
I enjoyed reading your interesting study, but there are a few things that need to be corrected.
- The introduction lacks a clear explanation of why readers should read this study, especially how it contributes to the literature.
- 3.1. The table is missing a number and title. The overall sample size looks small, especially in this table. Is it representative?
- For Study 2, it is a qualitative study using interviews. After using interviews, the authors should present the overall results in a single table.
- Finally, in Study 2, based on the overall interview results or in combination with Study 1, the authors should propose a single theoretical mechanism. If so, this should be justified with a theoretical rationale.
Author Response
|
Response to Reviewer 1 Comments |
||
|
1. Summary |
|
|
|
Thank you very much for taking the time to review this manuscript. Please find the detailed responses below and the corresponding revisions highlighted in the re-submitted files. |
||
|
2. Questions for General Evaluation |
Reviewer’s Evaluation |
Response and Revisions |
|
Does the introduction provide sufficient background and include all relevant references? |
Can be improved |
|
|
Is the research design appropriate? |
Can be improved |
|
|
Are the methods adequately described? |
Can be improved |
|
|
Are the results clearly presented? |
Can be improved |
|
|
Are the conclusions supported by the results? |
Can be improved |
|
|
3. Point-by-point response to Comments and Suggestions for Authors |
||
|
Comments 1: The introduction lacks a clear explanation of why readers should read this study, especially how it contributes to the literature. |
||
|
Response 1: Thank you for pointing this out. We have now emphasized the importance of the study and the lack of existing literature in this research area, particularly the absence of such studies in the Netherlands, at the end of the Introduction. Revision made in the manuscript in Paragraph 1 (Introduction), page 3, lines 109-111:
"This research area has limited existing literature, and in the Netherlands, there is currently no study exploring the perceptions, motivations, and influencing factors of restaurant owners, managers, and chefs regarding the implementation of healthier kids' menus."
Additionally, we have outlined an overarching exploratory research question (page 3, lines 113-115):
“Based on RECIM and previous literature, we explored the perceptions, attitudes, motivations, influencing factor and opportunities of Dutch restaurant owners, managers and chefs for implementing healthier kids menus [26].” |
||
|
Comments 2: 3.1. The table is missing a number and title. The overall sample size looks small, especially in this table. Is it representative? |
||
|
Response 2: We have added the number and title of the table in Section 3.1 line 248:
The total sample size consisted of 94 respondents. Given the challenges associated with reaching the target population, every response was considered valuable. While the sample may not be fully representative, it nevertheless offers important insights into a field that has not yet been extensively explored. In response to your comment, as well as the feedback from another reviewer, we have expanded our reflection on the sample size. This revision can be found in Paragraph 6.1, page 18, lines 645-654 of the manuscript:
“The relatively small sample size in Part I may have restricted the generalizability of our findings. Moreover, the contextual backdrop of pandemic-related disruptions could have influenced participants’ responses, introducing situational bias. However, each participation contributed to a better understanding in a novel research area. The insights will help to understand the motivations of the restaurants sector to ensure their cooperation in public health interventions. Giving these limitations, we recommend that future research should further explore perceptions within the restaurant sector to implement healthier kids menus using a larger, more representative sample, in order to validate both the findings and the measurement instruments, and identity temporal trends.” |
||
|
Comments 3: For Study 2, it is a qualitative study using interviews. After using interviews, the authors should present the overall results in a single table. |
||
|
Response 3: Thank you for your valuable comment. We have added a table summarizing the overall results of Part II of our study. As the table is relatively large, we refer to the Results of Part II in the manuscript in Paragraph 5.4, pages 14-16, line 529.
In addition, based on your comment and those of other reviewers—and in line with Part I—we have added a short concluding paragraph summarizing the key findings of Part II. The updated text can be found in the manuscript (Paragraph 5.5, page 16, lines 534-542):
“Part II identified several factors influencing the implementation of healthier kids' menus. Practical challenges included the increased time required for fresh ingredient preparation and difficulties in ensuring appropriate portion sizes for children. Commercial interest was hindering by low demand, infrequent sales, and limited menu space although a growing trend for vegetarian and organic options among young families was noted. Parental influence played a dual role by contributing both unhealthy choices and to an increased openness toward healthier options. Finally, innovative implementation strategies, such as creative presentation, integration with adult menus, and parental consultation, were highlighted as essential for facilitating change.” |
||
|
Comments 4: Finally, in Study 2, based on the overall interview results or in combination with Study 1, the authors should propose a single theoretical mechanism. If so, this should be justified with a theoretical rationale. |
||
|
Response 4: We added a theoretical mechanism to the manuscript and refer to the Retail Food Environment and Customer Interaction Model. The revised manuscript now includes the following explanation in the Introduction, page 2, lines 67-73:
“The Retail Food Environment and Customer Interaction Model (RECIM) [26], emphasizes a dynamic interplay between retailer-level factors (e.g., business operations, profit margins of restaurant owners), consumer behaviour (e.g., purchasing choices, preferences of parents and children), and broader contextual influences (e.g., food culture, policy, media) to achieve a shift towards implementing healthier kids’ menus. These interconnected elements shape food choices at the point of purchase and influence how and whether healthier kids menus are implemented in restaurants.”
A corresponding reflection to the RECIM and the Corporate Social Responsibility Model has been added to the Discussion, pages 16-18, lines 567-633:
‘’These findings align with the Retail Food Environment and Customer Interaction Model (RECIM) [26], suggesting that the shift towards implementing healthier kids’ menus is perceived as difficult due the financial and operational risks.’’ […] “While RECIM does not offer room for social responsibility, the Corporate Social Responsibility Model (CSRM) does [49], emphasizing that ethical and moral considerations can significantly influence corporate decision-making. In this context, the role of restaurants in improving the nutritional value of kids meals [29, 30] reflect a moral commitment to public health and social well-being, extending beyond purely economic or completive interest.” […] “This aligns with both RECIM and CSRM in recognizing how business models and product offerings are shaped by customer preferences, market positioning, and the pursuit of financial sustainability [26, 49].“ […] “These approaches align with RECIM, which emphasizes the dynamic interplay between consumer preferences and food retail practices [26].” […] “These strategies acknowledge how policy and economic context interact with retailer behaviour and consumer choices to shape the food environment [26].”
In addition, we have revised the Conclusion to better align our proposed practical solutions with the theoretical framework. This adjustment enhances the overall practical relevance of the study and reinforces the connection between theory and application. The revision in the Conclusion can be found on page 18, lines 663-669:
“The Retail Food Environment and Customer Interaction Model [26] provides a model to the strategies that focus on improving customer interactions and the influence of consumer preferences in shaping product offerings, as well as the role of business practices in meeting these demands. This approach provides practical starting points for integrating healthier kids menus into current restaurant operations. Notably, social responsibility appears to play a role in this process and should therefore be considered within theoretical economic approaches.” |
||
|
4. Response to Comments on the Quality of English Language |
||
|
Point 1: The English is fine and does not require any improvement. |
||
|
Response 1: Thank you for your positive feedback regarding the quality of the English language. We appreciate your assessment and are pleased to hear that no improvements are necessary. |
||
Reviewer 2 Report
Comments and Suggestions for Authors
Paper is very important, to improve the kwoledge about kids menus.
However the research is presented like 2 separated studies; and not as multi methos as shown in the title. Actually, paper presents 2 sections of methods, 2 sections of results... and this is not possible.
For this reason, the authors should reformulate the presentation of the paper.
Discussion also should be improved.
Author Response
|
Response to Reviewer 2 Comments
|
||
|
1. Summary |
|
|
|
Thank you very much for taking the time to review this manuscript. Please find the detailed responses below and the corresponding revisions highlighted in the re-submitted files. |
||
|
2. Questions for General Evaluation |
Reviewer’s Evaluation |
Response and Revisions |
|
Does the introduction provide sufficient background and include all relevant references? |
Can be improved |
|
|
Is the research design appropriate? |
Can be improved |
|
|
Are the methods adequately described? |
Can be improved |
|
|
Are the results clearly presented? |
Can be improved |
|
|
Are the conclusions supported by the results? |
Can be improved |
|
|
3. Point-by-point response to Comments and Suggestions for Authors |
||
|
Comments 1: Paper is very important, to improve the knowledge about kids menus. |
||
|
Response 1: Thank you for your kind comment. We’re glad to hear that you find the paper important and that it contributes to the understanding of kids’ menus. We appreciate your support. |
||
|
Comments 2: However, the research is presented like 2 separated studies; and not as multi methos as shown in the title. Actually, paper presents 2 sections of methods, 2 sections of results... and this is not possible. |
||
|
Response 2: We agree that this may have caused some confusion. Therefore, we have revised the structure and now present the study in two parts: Part I and Part II, instead of Study 1 and Study 2. This adjustment has been made both in the overall structure and throughout the text. An example of this updated text can be found on page 4, Paragraph 2.3, line 147. The Editor nor the other reviewers informed us that the journal does not accept this structure. |
||
|
Comments 3: For this reason, the authors should reformulate the presentation of the paper. |
||
|
Response 3: We have revised the structure accordingly and divided the study into two parts: Part I and Part II. |
||
|
Comments 4: Discussion also should be improved. |
||
|
Response 4: We have improved the discussion section in line with the suggestions from Reviewers 1, 3 and 4, by more clearly linking our findings to a relevant theoretical framework. Specifically, we now incorporate the Retail Food Environment and Customer Interaction Model and the Corporate Social Responsibility Model to interpret the perceived challenges associated with the adoption of healthier kids’ menus. The revised Discussion ( pages 16-18, lines 567-633):
‘’These findings align with the Retail Food Environment and Customer Interaction Model (RECIM) [26], suggesting that the shift towards implementing healthier kids’ menus is perceived as difficult due the financial and operational risks.’’ […] “While RECIM does not offer room for social responsibility, the Corporate Social Responsibility Model (CSRM) does [49], emphasizing that ethical and moral considerations can significantly influence corporate decision-making. In this context, the role of restaurants in improving the nutritional value of kids meals [29, 30] reflect a moral commitment to public health and social well-being, extending beyond purely economic or completive interest.” […] “This aligns with both RECIM and CSRM in recognizing how business models and product offerings are shaped by customer preferences, market positioning, and the pursuit of financial sustainability [26, 49].“ […] “These approaches align with RECIM, which emphasizes the dynamic interplay between consumer preferences and food retail practices [26].” […] “These strategies acknowledge how policy and economic context interact with retailer behaviour and consumer choices to shape the food environment [26].” |
||
|
4. Response to Comments on the Quality of English Language |
||
|
Point 1: The English is fine and does not require any improvement. |
||
|
Response 1: Thank you for your positive feedback regarding the quality of the English language. We appreciate your assessment and are pleased to hear that no improvements are necessary. |
||
Reviewer 3 Report
Comments and Suggestions for Authors
My recommendations are as follows:
Abstract: I recommend mentioning the role of the subjects by category out of the total of 94. I recommend mentioning that the questionnaire is not standardized. When mentioning the most relevant factors, I recommend mentioning the statistical significance values.
Introduction – I recommend mentioning the specific hypotheses for each of the two studies at the end of the section.
Line 136 the bibliographic index mentioned regarding the platform is not active and does not present any concrete information, I recommend clarifications.
Line 160 I recommend mentioning what 1, 5, 10 represent on the Likert scale.
3.2 Motivations for menu composition – I recommend that the information presented in table 2 not be duplicated when interpreting the results. I recommend rewriting.
Line 252 according to the related table, for the contributor reducing the presence of processed articles the value is 59.6 not 59.1, I recommend clarifications.
Study 2 recommends mentioning the main conclusions at the end of the study.
I recommend expanding the Discussion section by making new concrete correlations between the results of this study with results from previous studies targeting conditioning factors. I recommend mentioning the practical implications of this study.
Author Response
|
Response to Reviewer 3 Comments |
||
|
1. Summary |
|
|
|
Thank you very much for taking the time to review this manuscript. Please find the detailed responses below and the corresponding revisions highlighted in the re-submitted files. |
||
|
2. Questions for General Evaluation |
Reviewer’s Evaluation |
Response and Revisions |
|
Does the introduction provide sufficient background and include all relevant references? |
Can be improved |
|
|
Is the research design appropriate? |
Can be improved |
|
|
Are the methods adequately described? |
Can be improved |
|
|
Are the results clearly presented? |
Can be improved |
|
|
Are the conclusions supported by the results? |
Yes |
|
|
3. Point-by-point response to Comments and Suggestions for Authors |
||
|
Comments 1: Abstract: I recommend mentioning the role of the subjects by category out of the total of 94. I recommend mentioning that the questionnaire is not standardized. When mentioning the most relevant factors, I recommend mentioning the statistical significance values. |
||
|
Response 1: Thank you for pointing this out. We have specified the number of participants by their respective roles within the restaurant. In addition, we have clarified that the questionnaire used was unstandardized. These revisions can be found in the Abstract on page 1, lines 18-22. Updated text in manuscript:
“We used a mixed methods design in two consecutive study parts. Part I consisted of an online unstandardized questionnaire that was completed by 44 restaurant owners, 26 chefs, 18 managers, and 6 other restaurant employees (N = 94). This was followed by semi-structured interviews with 3 restaurant owners, 2 chefs, and 1 manager, to gather exploratory information in Part II (N = 6).” Thank you for your suggestion regarding the inclusion of statistical significance values in the abstract. We would like to clarify that the findings presented in the abstract are a synthesis of both quantitative data (Part I) and qualitative data (Part II). Given this integrative approach, it is not appropriate to report specific significance values in the abstract, as the results are not solely based on statistical testing but also include thematically analyzed qualitative findings. Including p-values in this context could be misleading and may not accurately reflect the nature of the combined results. However, statistical significance values are clearly reported and discussed in the main body of the manuscript, where appropriate. |
||
|
Comments 2: Introduction – I recommend mentioning the specific hypotheses for each of the two studies at the end of the section. |
||
|
Response 2: Thank you for your valuable comment. We partially agree with your observation. However, we did not formulate specific hypotheses, as our study was designed as an exploratory investigation. In line with the study design and following your comment and suggestions from other reviewers, we have now clearly articulated an overarching exploratory research question that reflects the objectives of both studies. This approach is more appropriate for the mixed methods design employed in our research. The revised manuscript now includes the following research question at the end of the Introduction (Paragraph 1, page 3, lines 113–115): “Based on RECIM and previous literature, we explored the perceptions, attitudes, motivations, influencing factors and opportunities of Dutch restaurant owners, managers and chefs for implementing healthier kids menus [26].” |
||
|
Comments 3: Line 136 the bibliographic index mentioned regarding the platform is not active and does not present any concrete information, I recommend clarifications. |
||
|
Response 3: Thank you for pointing this out. We have updated the bibliographic reference to ensure it links to an active and reliable source that provides concrete information about the platform. |
||
|
Comments 4: Line 160 I recommend mentioning what 1, 5, 10 represent on the Likert scale. |
||
|
Response 4: Thank you for pointing this out. We have added the labels for scores 1 and 10 in Paragraph 2.4.1, on page 4, lines 178-179:
“Additionally, participants offering kids’ menus were asked to rate the healthiness of the current kids’ menu on a 10-point Likert scale, ranging from 'Not healthy' (1) to 'Very healthy' (10).” .
In addition, we have applied this clarification elsewhere in the manuscript where Likert scales are mentioned. The revised text now reads:
|
||
|
Comments 5: 3.2 Motivations for menu composition – I recommend that the information presented in table 2 not be duplicated when interpreting the results. I recommend rewriting. |
||
|
Response 5: We agree with this comment and have revised the paragraph into a more concise summary of the table. The updated text now reads in Paragraph 3.2, on page 6, lines 258-263:
“All three restaurant groups were increasingly motivated to improve kids menus, driven by a rising demand for healthier options (49.4%). Key drivers were promoting innovation and originality in kids’ meals (48.1%), enhancing the restaurant’s public image (46.9%), and supporting a healthier lifestyle for families (43.2%). Additionally, many aim to introduce children to new flavors (39.5%) and make a meaningful contribution to children’s overall health (29.6%; see Table 2).” |
||
|
Comments 6: Line 252 according to the related table, for the contributor reducing the presence of processed articles the value is 59.6 not 59.1, I recommend clarifications. |
||
|
Response 6: We appreciate your attentiveness. We have made the correction in Paragraph 3.3, on page 7, line 274:
“According to the participants, the most important contributors to creating healthier kids’ menus include reducing the presence of processed items (59.6%), minimizing fried products (59.6%), lowering salt content (47.9%), and decreasing added sugars (46.8%; as shown in Table 3).” |
||
|
Comments 7: Study 2 recommends mentioning the main conclusions at the end of the study. |
||
|
Response 7: We agree with the comment to provide a clear summary for the reader. Accordingly, we have added a concluding paragraph (5.5 Conclusion Part II) at the end of Part II. The updated text now reads (lines 534-542):
“Part II identified several factors influencing the implementation of healthier kids' menus. Practical challenges included the increased time required for fresh ingredient preparation and difficulties in ensuring appropriate portion sizes for children. Commercial interest was hindering by low demand, infrequent sales, and limited menu space although a growing trend for vegetarian and organic options among young families was noted. Parental influence played a dual role by contributing both unhealthy choices and to an increased openness toward healthier options. Finally, innovative implementation strategies, such as creative presentation, integration with adult menus, and parental consultation, were highlighted as essential for facilitating change.” |
||
|
Comments 8: I recommend expanding the Discussion section by making new concrete correlations between the results of this study with results from previous studies targeting conditioning factors. |
||
|
Response 8: Thank you for your thoughtful comment. In response to comment 2 and other reviewers, we have incorporated a theoretical framework in the Introduction which is further discussed in the Discussion section with regard to our results.
The revised manuscript now includes the following explanation in the Introduction, page 2, lines 67-73:
“The Retail Food Environment and Customer Interaction Model (RECIM) [26], emphasizes a dynamic interplay between retailer-level factors (e.g., business operations, profit margins of restaurant owners), consumer behaviour (e.g., purchasing choices, preferences of parents and children), and broader contextual influences (e.g., food culture, policy, media) to achieve a shift towards implementing healthier kids’ menus. These interconnected elements shape food choices at the point of purchase and influence how and whether healthier kids menus are implemented in restaurants.”
A corresponding reflection to the RECIM and the Corporate Social Responsibility Model has been added to the Discussion, pages 16-18, lines 567-633:
‘’These findings align with the Retail Food Environment and Customer Interaction Model (RECIM) [26], suggesting that the shift towards implementing healthier kids’ menus is perceived as difficult due the financial and operational risks.’’ […] “While RECIM does not offer room for social responsibility, the Corporate Social Responsibility Model (CSRM) does [49], emphasizing that ethical and moral considerations can significantly influence corporate decision-making. In this context, the role of restaurants in improving the nutritional value of kids meals [29, 30] reflect a moral commitment to public health and social well-being, extending beyond purely economic or completive interest.” […] “This aligns with both RECIM and CSRM in recognizing how business models and product offerings are shaped by customer preferences, market positioning, and the pursuit of financial sustainability [26, 49].“ […] “These approaches align with RECIM, which emphasizes the dynamic interplay between consumer preferences and food retail practices [26].” […] “These strategies acknowledge how policy and economic context interact with retailer behaviour and consumer choices to shape the food environment [26].” |
||
|
Comments 9: I recommend mentioning the practical implications of this study. |
||
|
Response 9: We agree with the comment. Accordingly, we have added a sentence to the Conclusion Paragraph highlighting the added value and practical applicability of the proposed solutions from the manuscript. The revision in the Conclusion can be found on page 18, lines 668-674:
“The Retail Food Environment and Customer Interaction Model [26] provides a model to the strategies that focus on improving customer interactions and the influence of consumer preferences in shaping product offerings, as well as the role of business practices in meeting these demands. This approach provides practical starting points for integrating healthier kids menus into current restaurant operations. Notably, social responsibility appears to play a role in this process and should therefore be considered within theoretical economic approaches.” |
||
|
4. Response to Comments on the Quality of English Language |
||
|
Point 1: The English is fine and does not require any improvement. |
||
|
Response 1: Thank you for your positive feedback regarding the quality of the English language. We appreciate your assessment and are pleased to hear that no improvements are necessary. |
||
Reviewer 4 Report
Comments and Suggestions for Authors
The manuscript is well developed. Authors used a mixed methods design in two consecutive studies, consisting of an online questionnaire (n = 94) followed by semi-structured interviews to gather explanatory information (n = 6). The quantitative data were categorized into three groups: restaurants without kids menus (n = 18), restaurants with unhealthy kids menus (n = 24), and restaurants with (partially) healthy kids menus (n = 52). The authors should improve the theoretical framework by making the literature review more robust, more complete and more cohesive. The manuscript lacks (real and practical) contributions and the true contributions of the study in a pragmatic way. Authors discussed potential solutions to enhance demand and acceptance of healthier kids menus such as attractive names, storytelling, offering children's portions based on adult menus, and using participatory approaches in which parents, children and chefs co-create meal compositionWe recommend that the references at the end of the manuscript be improved. We suggest greater depth in the limitations of the study and lines of future research. We recommend that the references be mostly Scopus or WoS to make the research more consistent and of higher quality.
Author Response
|
Response to Reviewer 4 Comments |
||
|
1. Summary |
|
|
|
Thank you very much for taking the time to review this manuscript. Please find the detailed responses below and the corresponding revisions highlighted in the re-submitted files. |
||
|
2. Questions for General Evaluation |
Reviewer’s Evaluation |
Response and Revisions |
|
Does the introduction provide sufficient background and include all relevant references? |
Can be improved |
|
|
Is the research design appropriate? |
Can be improved |
|
|
Are the methods adequately described? |
Can be improved |
|
|
Are the results clearly presented? |
Can be improved |
|
|
Are the conclusions supported by the results? |
Can be improved |
|
|
3. Point-by-point response to Comments and Suggestions for Authors |
||
|
Comments 1: The manuscript is well developed. Authors used a mixed methods design in two consecutive studies, consisting of an online questionnaire (n = 94) followed by semi-structured interviews to gather explanatory information (n = 6). The quantitative data were categorized into three groups: restaurants without kids menus (n = 18), restaurants with unhealthy kids menus (n = 24), and restaurants with (partially) healthy kids menus (n = 52). |
||
|
Response 1: Thank you for your kind comment. We are glad to hear that you find the manuscript well developed. We appreciate your support. |
||
|
Comments 2: The authors should improve the theoretical framework by making the literature review more robust, more complete and more cohesive. |
||
|
Response 2: Thank you for your valuable comment. We fully agree with your observation. Based on prior literature and a relevant theoretical framework (Retail Food Environment and Customer Interaction Model), we improved the literature review. The revised manuscript now includes the following explanation in the Introduction, page 2, lines 67-73:
“The Retail Food Environment and Customer Interaction Model (RECIM) [26], emphasizes a dynamic interplay between retailer-level factors (e.g., business operations, profit margins of restaurant owners), consumer behaviour (e.g., purchasing choices, preferences of parents and children), and broader contextual influences (e.g., food culture, policy, media) to achieve a shift towards implementing healthier kids’ menus. These interconnected elements shape food choices at the point of purchase and influence how and whether healthier kids menus are implemented in restaurants.”
A corresponding reflection to the RECIM and the Corporate Social Responsibility Model has been added to the Discussion, pages 16-18, lines 567-633:
‘’These findings align with the Retail Food Environment and Customer Interaction Model (RECIM) [26], suggesting that the shift towards implementing healthier kids’ menus is perceived as difficult due the financial and operational risks.’’ […] “While RECIM does not offer room for social responsibility, the Corporate Social Responsibility Model (CSRM) does [49], emphasizing that ethical and moral considerations can significantly influence corporate decision-making. In this context, the role of restaurants in improving the nutritional value of kids meals [29, 30] reflect a moral commitment to public health and social well-being, extending beyond purely economic or completive interest.” […] “This aligns with both RECIM and CSRM in recognizing how business models and product offerings are shaped by customer preferences, market positioning, and the pursuit of financial sustainability [26, 49].“ […] “These approaches align with RECIM, which emphasizes the dynamic interplay between consumer preferences and food retail practices [26].” […] “These strategies acknowledge how policy and economic context interact with retailer behaviour and consumer choices to shape the food environment [26].” |
||
|
Comments 3: The manuscript lacks (real and practical) contributions and the true contributions of the study in a pragmatic way. |
||
|
Response 3: Our study adopted an exploratory approach, aimed at gaining a broader understanding of the topic and identifying key factors that may influence future research and practice. Based on your feedback, as well as suggestions from other reviewers, we have strengthened the theoretical foundation of our study by integrating a theoretical framework. Based on the theoretical framework and previous literature, we also formulated a research question. In the Introduction, we have incorporated a theoretical framework which is further discussed in the Discussion section with regard to our results.
The revised manuscript now includes an overarching exploratory research question (page 3, lines 113-115):
“Based on RECIM and previous literature, we explored the perceptions, attitudes, motivations, influencing factor and opportunities of Dutch restaurant owners, managers and chefs for implementing healthier kids menus [26].”
As indicated in response 2, the revision in the Introduction can be found on page 2, lines 67-73.
As indicated in response 2, the revision in the Discussion can be found on pages 16-18, lines 567-633.
In addition, we have revised the Conclusion to better align our proposed practical solutions with the theoretical framework. This adjustment enhances the overall practical relevance of the study and reinforces the connection between theory and application. The revision in the Conclusion can be found on page 18, lines 668-674:
“The Retail Food Environment and Customer Interaction Model [26] provides a model to the strategies that focus on improving customer interactions and the influence of consumer preferences in shaping product offerings, as well as the role of business practices in meeting these demands. This approach provides practical starting points for integrating healthier kids menus into current restaurant operations. Notably, social responsibility appears to play a role in this process and should therefore be considered within theoretical economic approaches.” |
||
|
Comments 4: Authors discussed potential solutions to enhance demand and acceptance of healthier kids menus such as attractive names, storytelling, offering children's portions based on adult menus, and using participatory approaches in which parents, children and chefs co-create meal composition. |
||
|
Response 4: Thank you for your kind comment. We appreciate your support. |
||
|
Comments 5: We recommend that the references at the end of the manuscript be improved. We recommend that the references be mostly Scopus or WoS to make the research more consistent and of higher quality. |
||
|
Response 5: Thank you for your comment. We would like to clarify that this is a relatively new and still limited area of research. Therefore, we deliberately chose a broad search strategy to gather as much relevant literature as possible. This approach allowed us to include a wide range of studies that contribute to understanding the topic, which we believe enhances the comprehensiveness and relevance of our study. This approach enabled us to include a wide range of studies that contribute to a deeper understanding of the topic, thereby enhancing the scientific foundation and relevance of our study. |
||
|
Comments 6: We suggest greater depth in the limitations of the study and lines of future research. |
||
|
Response 6: We agree with your comment. Therefore, we have expanded our reflection on the sample size and translated this into concrete recommendations for future research. This revision can be found in Paragraph 6.1, page 18, lines 645-654 of the manuscript.
Updated text in the manuscript: “The relatively small sample size in Part I may have restricted the generalizability of our findings. Moreover, the contextual backdrop of pandemic-related disruptions could have influenced participants’ responses, introducing situational bias. However, each participation contributed to a better understanding in a novel research area. The insights will help to understand the motivations of the restaurants sector to ensure their cooperation in public health interventions. Giving these limitations, we recommend that future research should further explore perceptions within the restaurant sector to implement healthier kids menus using a larger, more representative sample, in order to validate both the findings and the measurement instruments, and identity temporal trends.” |
||
|
4. Response to Comments on the Quality of English Language |
||
|
Point 1: The English is fine and does not require any improvement. |
||
|
Response 1: Thank you for your positive feedback regarding the quality of the English language. We appreciate your assessment and are pleased to hear that no improvements are necessary. |
||
Round 2
Reviewer 1 Report
Comments and Suggestions for Authors
The revision is now acceptable.
Well done.
Reviewer 2 Report
Comments and Suggestions for Authors
I consider that the correction improve the paper.
About the structure I will like that editor argue about this.
Reviewer 3 Report
Comments and Suggestions for Authors
No comments